# Novel hormonal therapy versus standard of care—A registry-based comparative effectiveness evaluation for mCRPC-patients

**Paulina Jonéus**[1⊚], **Per Johansson**[1,2,3⊚], **Sophie Langenskiöld**[ID][2,4⊚]*

**1** Department of Statistics, Uppsala University, Uppsala, Sweden, **2** Centre for Health-Economic Research, Uppsala University, Uppsala, Sweden, **3** YMSC, Tsinghua University, Beijing, China, **4** Department of Medical Sciences, Uppsala University, Uppsala, Sweden

⊚ These authors contributed equally to this work.
* Sophie.langenskiold@medsci.uu.se

## Abstract

### Background

This paper presents results from one of the few comparative effectiveness evaluations of novel antiandrogen medications (NHT) against standard of care (SoC) for patients suffering from metastatic castrate-resistant prostate cancer (mCRPC).

### Methods

The design and the analysis are published in a protocol before accessing outcome data. Two groups of patients are balanced on hundreds of important covariates measured before the prostate cancer diagnosis and up to the date of the prescription. While the design yields balance on the observed covariates, one cannot discard the possibility that unobserved confounders are not balanced. The unconfoundedness assumption is assessed by estimating placebo regressions on two health measures, not included in the design but added together with the outcome data after protocol publication.

### Results

We find a substantial (64 percent) increase in mortality for patients prescribed with NHT rather than SoC. However, based on the results from one of the two placebo regressions, we cannot rule out that the difference in mortality may be due to confounding. Using a bounding strategy of the effect, we can, however, rule out that NHT reduces mortality compared to SoC. Under an empirical valid assumption that most mCRPC patients who die suffer from bone metastases, we have a strong indication of increased skeleton-related events in patients if prescribed NHT against SoC.

### Conclusions

Generally, the SoC for this group of patients is docetaxel. Given the substantially higher costs of many of the NHT, the finding of no positive effects from NHT on both mortality and

**Data Availability Statement:** According to Swedish legislation (https://etikprovningsmyndigheten.se/for-forskare/vadsager-lagen/) data cannot be made available for

use beyond what has been approved by the ethical review board. Therefore, the data cannot be made publicly available. However, anyone can request data from National Bureau of Health and Welfare, Statistics Sweden, or National Prostate Cancer Register after having received an ethical approval for the research. Contact mikrodata@scb.se, Registerservice@socialstyrelsen.se, and datauttag-rcc@rccmellan.se for questions regarding the respective datasets, and registrator@etikprovning.se for questions regarding the ethical permission which is a prerequisite for a datarequest.

**Funding:** Our research was funded by the Dental and Pharmaceutical Benefits Agency (grant number 02823/2017) received by PhD Sophie Langenskiöld. The funder had no role in the study design, data collection, and statistical analyses. Neither did they influence the decision to publish or the preparation of the manuscript.

**Competing interests:** The authors have declared that no competing interests exist.

SRE is important. More comparative studies, including studies analysing quality of life outcomes, are thus needed.

## Introduction

Prostate cancer (PC) is the fifth leading cause of cancer death in men worldwide, and almost all mortalities arise when patients have progressed to the advanced stage: metastatic castrate-resistant prostate cancer (mCRPC) [1]. Various treatment alternatives are available for these patients, and in the last two decades, chemotherapy and new hormonal treatment (NHT) medications have revolutionized treatment [2–7]. However, our structured literature review found only two studies comparing chemotherapy and NHT in terms of survival [8, 9] None of the studies finds a significant difference, however, it's unclear of the unbiasedness of these estimates. In Chowdhury et al., [9], for example, the patients on docetaxel are more progressed at baseline than those on NHT, and the adjustments for these differences are only briefly mentioned in the paper.

This paper presents the results from a comparative effectiveness evaluation of two of these NHTs, abiraterone acetate (AA) in combination with prednisone and enzalutamide (ENZ), against standard of care (SoC). In our analysis, we estimate the average treatment effect on overall mortality. In addition, we estimate the effect on the prevalence of two secondary outcomes, skeleton-related events (SRE) and pain.

Data are collected from population registers administrated by the National Board of Health and Welfare (NBHW) and Statistics Sweden (SCB). The population is restricted to all men in the NBHW register with a PC diagnosis before 2017. The maintained assumption of the study design is that given observed pre-treatment covariates, the treatments (NHT or SoC) are unconfounded. That is, we assume we observe the relevant covariates associated with the respective outcomes and, on the same time, determine the prescription. Mortality data from NBHW was added after the publication of the protocol; the protocol, including specified details concerning the design and analyses, is found in Jonéus et al. [10].

Real-world data is increasingly accepted as evidence in the regulatory process (see, e.g., [11–15]). A relevant question is, therefore, whether or not a protocol similar to the requirement for a randomized control trial can be pre-published. As yet, there is no consensus on the content requirement of such a protocol [16]. As a potential input to the discussion, this paper illustrates that an observational study based on a pre-published protocol can encapsulate the same level of detail as a protocol for a randomized experiment.

The analysis concerns NHT usage in clinical practice from June 2015 to June 2018, which corresponds to the period when these drugs were reimbursed for mCRPC patients who had failed androgen deprivation therapy (ADT) and were not yet suited for docetaxel (pre-chemotherapy), or for patients who had failed docetaxel (post-chemotherapy). In June 2018, AA was additionally reimbursed as an add-on for high-risk metastatic hormone-sensitive prostate cancer (mHSPC) patients unsuited for docetaxel. Chemotherapy with docetaxel is found in previous, yet unpublished, qualitative work to be the SoC for this group of patients. Since docetaxel was approved in 2004 and there was no other first-line treatment for mCRPC until the introduction of the NHTs, this creates a unique situation for estimating the effect of the new NHT against SoC using historical data.

While the design yields balance on the observed covariates, one cannot discard the possibility that unobserved confounders are not balanced. Therefore, data from the national PC

register (NPCR) was added *after* protocol publication, allowing us to objectively assess the unconfoundedness assumption using placebo regressions on two covariates measured before the NHT prescription. The PSA levels and the Gleason score are the two covariates judged by specialists as potential confounders.

We find a substantial increase in mortality for patients prescribed NHT rather than SoC. No differences in the morbidity outcomes were found. Based on the results from the placebo regression on the Gleason score, we cannot rule out that the differences in mortality are due to confounding. However, an extended analysis allows ruling out that the estimates on mortality would have been negative and statistically significant in the presence of confounding, i.e., the opposite effects. Thus, this analysis allows us to rule out a reduction in mortality for patients prescribed NHT instead of SoC.

The remainder of the paper discusses the data and covariates in section 2. Section 3 provides a discussion of the design in Jonéus et al. [10], together with a description of the analysis sample and the outcomes. Section 4 sets up the analysis model, and section 5 presents the results. The sensitivity analyses are conducted in section 6. Finally, the paper concludes in section 7.

## Methods

Rich data concerning all patients' prostate cancer and disease progression is collected from the NBHW registers. The inpatient care register contains information about patients' admission and discharge dates and information on the principal and secondary diagnoses for hospitalizations (using the ICD 10 classifications). The cancer register contains data on prostate cancer diagnosis. The pharmaceutical register contains the date of prescription and dispensing of drugs and their ATC class. Based on this information, we restrict the population to all men in Sweden with a PC diagnosis from 1986 to 2016.

The SCB data includes information on age, marital status, educational level, and country of birth, as well as a large set of covariates measuring security benefits, such as pensions, income, and sick leave. We collect data on socio-economic status at the year of diagnosis and the two preceding years. In a few cases of missing values, the average of the information from previous years is used for each individual. In addition, information on educational level is sometimes missing; in these cases, a five-nearest-neighbour approach is used to impute missing values. The study was approved by the Regional Ethical Review Board in Uppsala, Sweden (Dnr 2017/482).

## Sample restrictions

The fact that the two NHTs were not offered by Swedish healthcare before being subsidized provides a unique opportunity for evaluating NHT against prior SoC in clinical practice. However, (i) the comparison population should be as similar as possible to those prescribed a NHT, and (ii) the two populations should be prescribed their treatment as 1st line treatment and have been provided with the same quality of healthcare. Finally, (iii) the outcome of the comparisons should not be affected by later being prescribed an NHT. The first requirement is solved by adjusting for many differences in disease progression using the detailed register data, which will be discussed in the next section. The second requirement is solved by sampling the comparisons with a PC diagnosis close in calendar time to those prescribed NHT. The last requirement is solved by not following comparisons after NHT became subsidized, i.e., until June 2015 at the latest.

To provide an intuition for the consequences of the third requirement, consider a patient with a diagnosis in June 2010. This patient could, at the earliest, be given an NHT in June 2015

after a waiting time (WT) of a maximum of five years. For that reason, this patient could only be among the comparisons until June 2015. Any longer follow-up time entails the risk that the outcomes are affected by a latter prescription of an NTH. Note that with this sampling, the potential follow-up time will be shorter than five years as most patients are not prescribed SoC at the time of their diagnosis. For example, for NHT, the majority were prescribed an NHT after more than 12 months of WT and none was prescribed an NHT at the time of diagnosis.

In order to obtain a sufficiently large group of comparison patients, we sampled all men with PC diagnosis from June 1, 2008, to June 1, 2010, which gave us a sample of 19,456 patients.

Sampling restrictions on the comparisons given by requirements (ii) and (iii) restrict, by definition, the NHT sample to patients prescribed an NHT fairly short after the diagnosis. We decided to sample those diagnosed from June 2012 to June 2015 and prescribed an NHT within 36 months after diagnosis. With these restrictions, the treated population consists of 1,285 NHT patients. As we will have mortality data until June 2020, we have a minimum follow-up period of two years and a maximum follow-up period of five years. For further details on the sampling, see Jonéus et al. [10].

Only a few patients were prescribed an NHT within six months of diagnosis; most were prescribed it after more than 12 months. As no patients received treatment less than four months after diagnosis, the data can be described as a stratified sample with 33 strata, $w = 4, . . ., 36$ months (see S1 Fig). Therefore, the analysis design considers each stratum separately.

Descriptive statistics on a sub-set of pre-diagnosis covariates for the two groups are displayed in Table 1. From the table, we can see that those prescribed NHT are older and have worse health than the comparison group, especially at the time of diagnosis, but also, to some extent, 12 months before. The Elixhouser comorbidity index is based on ICD diagnostic codes

**Table 1. Descriptive statistics for a sub-set of pre-diagnosis covariates: Standard of care (SoC) and Novel hormonal therapy (NHT) patients and the mean difference (Diff) between the two groups.**

| Description | SoC Mean (stdev) | NHT Mean (stdev) | Diff. Mean |
|---|---|---|---|
| Age at diagnosis | 69.40 (9.16) | 70.85 (8.45) | -1.45*** |
| # Hospital visits 1 month bf. diagnosis | 0.47 (0.84) | 0.71 (1.10) | -0.23*** |
| # Hospital visits 1–6 months bf. diagnosis | 1.38 (2.36) | 1.68 (3.15) | -0.30*** |
| # Hospital visits 6–12 months bf. diagnosis | 0.83 (1.96) | 0.82 (2.60) | 0.01 |
| # Hospital visits 1–60 months bf. diagnosis | 7.12 (11.00) | 7.85 (23.72) | -0.73 |
| Elixhouser index = 0, at diagnosis | 0.53 (0.50) | 0.44 (0.50) | 0.09*** |
| Elixhouser index = 1–4, at diagnosis | 0.43 (0.50) | 0.51 (0.50) | -0.08*** |
| Elixhouser index ≥5, at diagnosis | 0.03 (0.18) | 0.05 (0.22) | -0.02*** |
| Elixhouser index = 0, 12 months bf. diagnosis | 0.62 (0.49) | 0.54 (0.50) | 0.08*** |
| Elixhouser index = 1–4, 12 months bf. diagnosis | 0.36 (0.48) | 0.43 (0.50) | -0.07*** |
| Elixhouser index ≥5, 12 months bf. diagnosis | 0.02 (0.13) | 0.03 (0.17) | -0.01** |
| Less than secondary school education | 0.38 (0.49) | 0.36 (0.48) | 0.02 |
| Secondary school education | 0.39 (0.49) | 0.39 (0.49) | -0.01 |
| Living with a partner | 0.65 (0.48) | 0.64 (0.48) | 0.01 |
| Born in the Nordic countries | 0.95 (0.22) | 0.94 (0.23) | 0.01 |
| Number of individuals | 19,456 | 1,285 | |

Means (standard deviations)

*p <0.1

**p <0.05

***p <0.01.

and includes 31 comorbidities, which can be summarized in a single numeric score [17] Comorbidities have been shown to influence cancer survival, and the Elixhauser index is found to be a good comorbidity risk adjustment method [18]. We follow Menendez et al. [19] and categorize each comorbidity score into one of four index groups where an index of 0 implies no comorbidities and an index of $\geq 5$ implies several comorbidities and thereby the worst health condition. As seen in Table 1, the Elixhouser index indicates a larger difference in comorbidity status at the time of the diagnosis, where NHT patients seem to have a worse health status than when measured 12 months before diagnosis. The number of hospital visits (# hospital visits) shows a statistically significant difference in the period just before diagnosis, but no difference between six and 60 months before diagnosis. In short, there is a clear indication of more severe health progression in the NHT treatment group than in the comparison group before PC diagnosis.

## Variables measuring sickness progression after diagnosis

Six variables are used to measure sickness progression each month after the diagnosis until the month of treatment for those prescribed an NHT and for 36 months for the SoC patients [10]. The latter is because we do not have access to the date regarding when SoC patients were administered SoC. However, information on these variables is only used up to the potential time when patients could serve as controls to patients on NHT, i.e., a maximum of 36 months after PC diagnosis or until the time of death (for more details, see Jonéus et al., [10]). The variables are (i) the cumulative number of hospital visits, (ii) presence of metastases (ICD C77-C79), (iii) number of months with visceral metastases (ICD C78), (iv) number of months with skeleton metastases (ICD C795), (v) total number of collected daily doses of any antiandrogen, GnRH-agonist or GnRH-analog3, and (vi) the indicator of being prescribed only Bicalutamide or Bicalutamide+GnRH up until a specific month.

## Design and outcome data

### Design

An often-used strategy in designing a study is to base it on the estimated propensity score and to find matches based on this scalar. This means that with respect to this estimated propensity score, the control individual can be chosen by finding the most similar treated individual in month $t$ to form a pair. The propensity score matching strategy can be time-consuming as the final specification of the propensity score needs to be determined iteratively until we have obtained balance in the covariates between the two groups. In this paper, we instead make use of the entropy balancing scheme, or algorithm, suggested by Hainmueller [20] for balancing the covariates between the two groups. The algorithm starts by assuming that the covariates are perfectly balanced between the two groups (i.e. uniform base weights). If the two groups are not balanced, the algorithm finds the scalar weight to each treated unit ensuring that the reweighted comparison groups have the same mean (and higher moments) of the included covariates as the treated group. Then, the weights are recalibrated until the pre-defined balance across the groups is achieved (for more details, see Jonéus et al. [10]). In order to adjust for the sickness progression from the time of diagnosis to prescription, the balancing is performed separately for each of the 33 strata. Thus, for each stratum, we adjust for covariates measured at the time of diagnosis and for the stratum-specific covariates capturing the sickness progression.

The complete set of covariates includes interactions and second-order terms, in addition to the linear terms. In addition to balancing the means, we also balance the variances of the main covariates between the two groups. This results in a balance of up to 56 covariates, depending on the stratum.

## Outcome data

We have one primary and two secondary morbidity outcomes. The primary outcome is all-cause mortality (DEAD); the two secondary outcomes are SRE and PAIN. SRE is an indicator of a skeleton-related event, and PAIN is an indicator of severe pain. The evaluation period is 24 months after treatment.

DEAD is an indicator variable of zero if the patient is alive at the end of each 30 days and one if the patient died during that specific period. Patients are assumed to suffer a skeleton-related event (SRE) if they experience a hospitalization because of pathologic fracture (ATC codes M485, M495, M844, and M907) or spinal cord compression (G550, G834, G952, G958, G959, and G992) [21]. The SRE indicator is one for every 30 days of such hospitalizations, and zero for other periods.

Patients are assumed to suffer severe pain if they receive prescriptions for neuropathic pain, i.e., opiates in combination with tramadol and paracetamol (ATC-codes N02AA, N02AX02, and N02BE01). The PAIN indicator is one for each 30-day period in which the patient has received such a prescription, and zero for the other periods.

Mortality data were added after the publication of the pre-analysis plan (10). In this data, 9,389 of the 20,757 patients (45%) in the sample have a date of death. Of these 9,389 patients, 3,832 have the PC diagnosis registered as the main cause of death (i.e., ICD C61).

## Analysis and estimation

Let $Y_{i,m}$ denote any of the three outcomes observed for patient $i$ in month $m$. Let $T_i = 1$ if the patient is prescribed NHT and $T_i = 0$ otherwise. We regress $Y_{i,m}$ on $T_i$ and stratum indicators (see S2 Text for details). The monthly effects are estimated using the entropy balancing weights, and standard errors are estimated using a robust covariance estimator (see, e.g., Huber, [22]). The estimated confidence intervals are adjusted for multiple testing using Bonferroni correction, and we let the overall significance level be 5%. With one primary and two secondary outcomes, the significance level of the single outcome is 1.67% ( = 100*0.05/3).

Note that every comparison individual has a weight in each of the 33 strata, that is, the month after diagnosis when the NHT patients were prescribed the drug. If we had data on mortality when designing the study, SoC patients who died any time before month 36 would have been removed from consideration. Around 8% of the comparisons in each stratum have died before, or in, this monthly stratum. The entropy weights were recalculated after censoring patients in the comparison group at the time of death. As the results using these new entropy weights are all qualitatively similar (see S2, S8 and S9 Figs) to the results based on the pre-specified design, we present the result with the original design below for transparency. All Analyses and estimations are conducted with R version 3.5.1.

## Results

The mortality analysis results are presented in sub-section 5.1, and the morbidity outcomes are given in sub-section 5.2. Finally, the last sub-section presents results from the separate analysis of cases with prescriptions of NHT given early or late after diagnosis.

### Mortality

The results on the incidence of mortality for the 24 periods of 30 days are presented in Fig 1, while effects from a survival analysis are presented in Fig 2. In the survival analysis, we estimate a discrete time Cox regression model. For the discrete-time Cox regression model and the

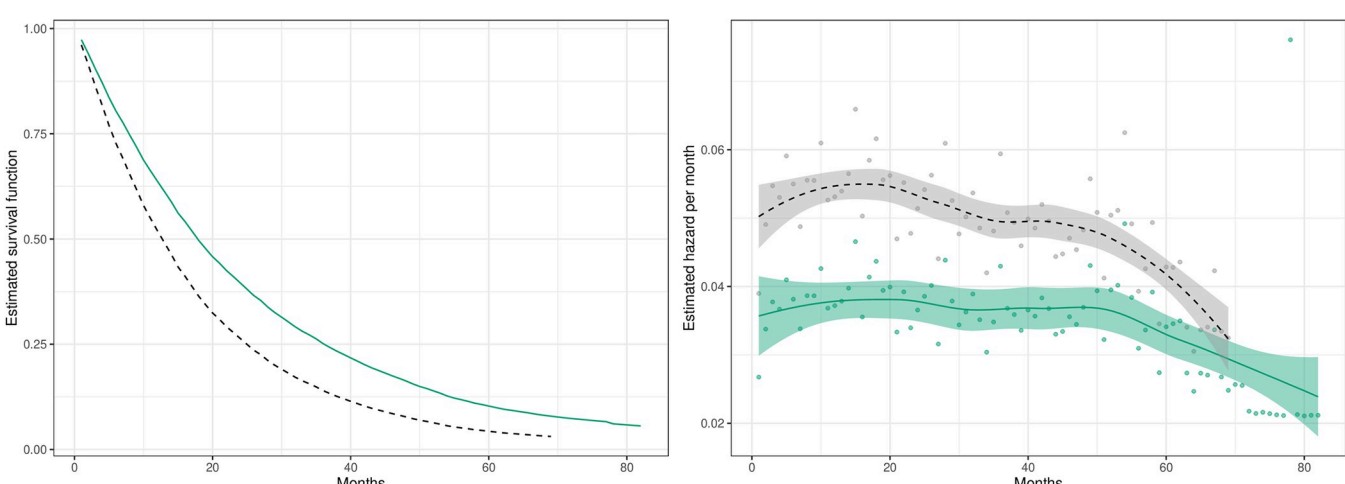

**Fig 1. The effect on mortality of NHT against SoC.** Estimates, 95% Bonferroni confidence intervals, and overall mortality in each month (x).

**Fig 2. The estimated survival function and conditional probability of dying each month (i.e., the monthly hazard) for NHT (−−) and SoC (-—-) patients up to 82 months after a potential drug prescription of NHT.**

results from the analysis, see S1 Table. In this analysis, we have a follow-up period of 69 months for NHT patients and 82 for SoC patients.

Fig 1 displays the point estimates, the 95% Bonferroni corrected confidence intervals, and overall mean mortality on 24 periods of 30 days, in the following denoted months. A clear pattern of higher mortality is seen in patients if prescribed NHT against SoC. Further, the 95% confidence intervals do not cover zero 16 times out of 24. The average increase in mortality over the 24 months is 2.3 percentage points, or 64% when normalizing with the overall mean mortality for the two groups (0.0357).

The estimated survival functions for the two groups is displayed in Fig 2A, while the corresponding hazard functions, with confidence intervals, are given in Fig 2B. The figure show that the monthly hazard is around 5 percent for the NHT patients and around 3.5% for SoC. These results confirm those from the incidence analysis: a substantial difference in mortality in months 8 to 24. However, from this graph, we can also see that this difference persists for 60 months.

## Skeleton-related events and severe pain

We observed 760 unique patients (4%) suffering from an SRE, 12% in the NHT group, and 3% in the SoC group. Only 376 (2%) unique patients have severe pain, according to our definition of PAIN, and in the group of NHT patients, the proportion is 0.2%.

However, we only observed an indication of an SRE or PAIN when the patient was alive. As the mortality is higher for the NHT group, we are consequently more likely to observe any of these morbidity outcomes for SoC patients than for NHT patients. To manage this problem, we estimate the bounds of potential effects. First, we let all patients who die have either a no morbidity or a morbidity outcome. For the SRE, this means letting SRE = 0 or SRE = 1. Then, since the mortality with NHT is observed to be higher than with SoC, the first case with SRE = 0 provides a lower bound estimate of the effectiveness of NHT versus SoC in preventing SRE, while the second with SRE = 1 one provides an upper bound.

The results from these analyses on SRE are presented in Fig 3. We find no differences in effect on SRE if we treat all who died as not suffering from an SRE. On the other hand, if all who died suffered from an SRE, then the NHT patients would have had less SRE if they had instead had SoC. According to Wong et al. [23], most mCRPC patients who die suffer from

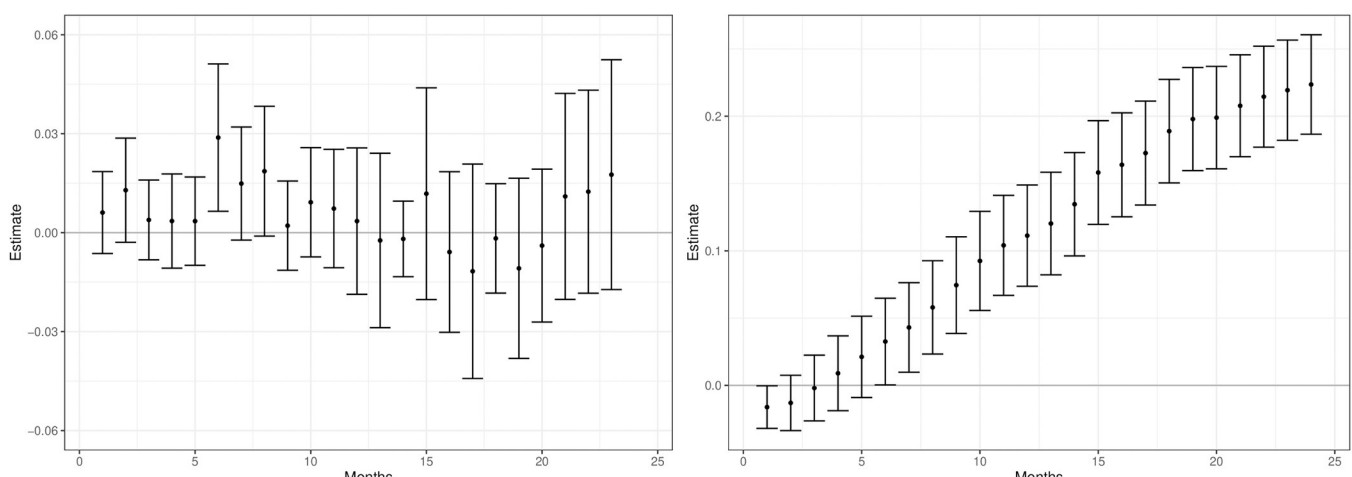

**Fig 3. Effects on SRE of NHT against SoC.** Estimates and 95% Bonferroni corrected confidence intervals. Lower (left panel) and upper (right panel) bounds of potential effect.

bone metastases. As SRE is a complication of bone metastases, it is possible that the 'true' differences are closer to the upper bound results than the lower.

As only 2% of the patients have severe pain and the proportion is only 0.2% among NHT patients, the effects are very imprecisely estimated. Nevertheless, for transparency, the results from these analyses are presented in S3 and S4 Figs.

## Effects on early and late prescription

We do the analysis presented in Section 5.2 separately for those with less than the median time (less than 20 months) to NHT prescription from PC diagnosis, and longer or equal to the median time (more than or equal to 20 months) on the three outcomes. The results indicate that the negative effect could be substantially greater for those with early rather than late prescriptions. However, as almost all confidence intervals overlap zero, no firm conclusion can be made (see S5 Fig). The results for the two morbidity outcomes did not show any clear pattern but are included for transparency (see S6 and S7 Figs).

## Summary of the results

Analysis of both death incidence and survival time provides a coherent pattern of higher mortality in patients if prescribed NHT against SoC. Under the valid assumption (cf. Wong et al. [23]) that most mCRPC patients who die suffer from bone metastases, we have a strong indication of increased skeleton-related events in patients if prescribed NHT against SoC. There is an indication that the increased mortality for patients being prescribed NHT is primarily for those with early rather than late prescriptions.

## Sensitivity analyses

We also matched data from NPCR to the analysis sample when adding mortality data. These data contain more detailed information on patients' health with regard to the PC. Three covariates, judged by specialists to be important confounders and measured at the time of PC diagnosis, are used: the PSA levels (SPSA), Gleason score (GleasSa), and metastases (Mstad).

The validity of the design is assessed by estimating 'placebo' effects on these covariates. If there are statistically significant effects on these covariates, available data from the population register may not be sufficient to control for confounding bias.

One crux with this analysis is that the NPCR does not have full coverage, and there are partly missing data on the covariates. Therefore, we analyse the missing data problem in the following sub-section before turning to the placebo regression.

## Missing data

There are 1,083, 1,367 and 14,923 missing observations for SPSA, GleasSa and Mstad, respectively. The placebo regressions are valid if the missing observation can be treated as missing at random. From the analysis of missingness (see S1 Text), we conclude that Mstad is not a valid proxy outcome. GelasSa is, however, a valid proxy outcome, as we can treat the observations as missing at random except for stratum 4, 5, and 26. As the degree of missingness for SPSA and remaining strata for GleasSa are low, we remove patients with missing observations from the sensitivity analyses. It should be acknowledged that the conclusion that the data is missing at random may be a false negative. Thus, the following analysis needs to be interpreted with this possibility in mind.

## Placebo regression

We use the same analysis as in the main analysis but instead regress the two proxy outcomes, i.e., SPSA or GleasSa, on the NHT indicator. As the entropy weights are at the stratum level, we estimate the association at the stratum level (see S2 Text) This means we get 33 and 30 (as we exclude stratum 4, 5, and 26) estimates of the association for the SPSA and GleasSa, respectively. The proxy outcomes are standardized to have zero mean and unit variance per strata. The results from these analyses are shown in Fig 4.

The left panel shows the results for SPSA, and the right the results for GleasSa. There are no differences between the two groups with regard to SPSA. However, NHT patients are seen to have between 0.27 to 0.96 standard deviation higher levels on the GleasSa than SoC patients. Most of these estimates are also statistically significant. S10 Fig reports qualitatively similar results from the placebo regressions with the updated weights.

The Gleason score is a confounder in the analysis if it is also associated with the outcome, conditional on the covariates. To this end, we recalculate the weights, including GleasSa. The results from this analysis are presented in S11 Fig, and are similar to the results from the main analysis. All estimates are positive, with one exception for the first month, and the majority of these are statistically significant. The overall picture of increased mortality from NHT is thus not changed due to the imbalance in the GleasSa.

There could, of course, be other confounders pointing to why a firm conclusion of increased mortality from being prescribed NHT could be questioned. Given the differences in cost of the two types of treatment, it is, however, interesting to consider whether we can rule out a reverse effect on mortality for *each stratum*. The substantial imbalance of 0.96 standard deviations found for stratum six is used in this derivation. S2 Text details a framework for this analysis, but the idea and results from the analysis are provided in the following two paragraphs.

The idea is as follows: using the imbalance of 0.96 standard deviations, we derive the smallest possible association an unobserved confounder needs to have with mortality to obtain a statistically significant and reversed sign of the mortality estimate. We then compare this derived association with the estimated associations between (i) mortality and age, and (ii) mortality and number of months with skeleton metastases (metastases) in the *unadjusted sample*. If the estimated association between mortality and these two variables is larger than the derived

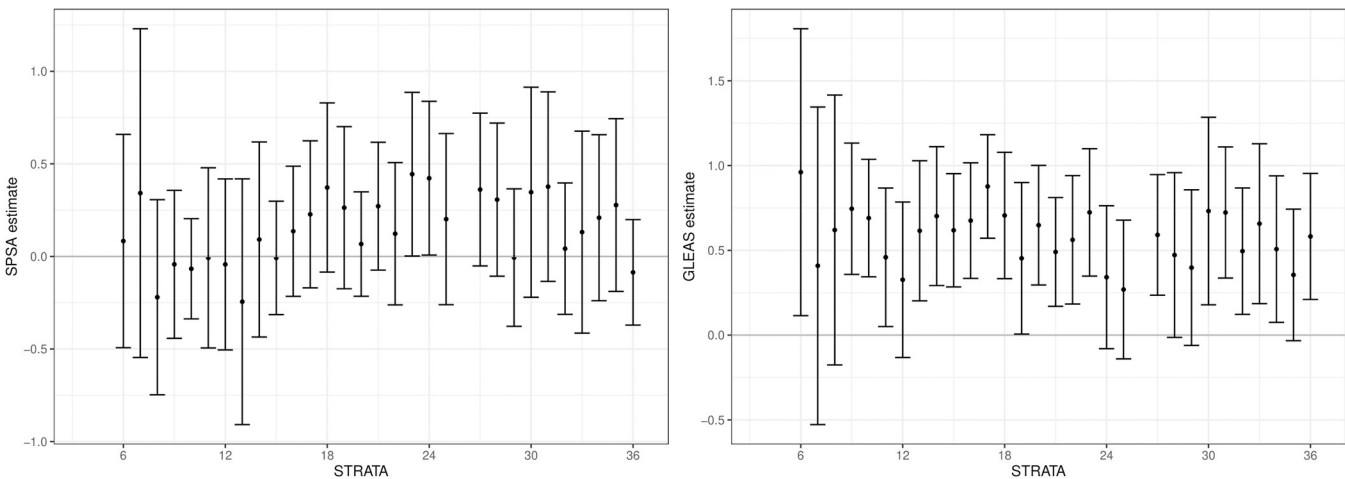

**Fig 4.** Results from the placebo regression on SPSA (left) and Gleason (right). Estimate and 95% Bonferroni corrected confidence interval for each month.

**Table 2. Results from the sensitivity analyses.**

| Month | Lower limit $(\hat{\delta}_m^l)$ | Estimates | |
|---|---|---|---|
| | | **Metastases** | **Age** |
| 1 | 0.0054 | 0.00079 (0.0013) | 0.00031 (0.00042) |
| 2 | 0.0210 | 0.00205 (0.0013) | 0.00034 (0.00037) |
| 3 | 0.0273 | 0.00166 (0.0012) | 0.00144 (0.00040) |
| 4 | 0.0292 | 0.00225 (0.0013) | 0.00188 (0.00040) |
| 5 | 0.0369 | 0.00212 (0.0013) | 0.00123 (0.00039) |
| 6 | 0.0304 | 0.00279 (0.0013) | 0.00183 (0.00037) |
| 7 | 0.0265 | 0.00225 (0.0013) | 0.00163 (0.00037) |
| 8 | 0.0427 | 0.00362 (0.0013) | 0.00201 (0.00039) |
| 9 | 0.0488 | 0.00336 (0.0013) | 0.00229 (0.00038) |
| 10 | 0.0511 | 0.00284 (0.0014) | 0.00202 (0.00039) |
| 11 | 0.0381 | 0.00339 (0.0013) | 0.00205 (0.00039) |
| 12 | 0.0428 | 0.00226 (0.0012) | 0.00156 (0.00036) |
| 13 | 0.0528 | 0.00334 (0.0014) | 0.00149 (0.00036) |
| 14 | 0.0500 | 0.00452 (0.0014) | 0.00248 (0.00041) |
| 15 | 0.0722 | 0.00380 (0.0014) | 0.00204 (0.00039) |
| 16 | 0.0423 | 0.00390 (0.0013) | 0.00176 (0.00038) |
| 17 | 0.0650 | 0.00419 (0.0014) | 0.00158 (0.00038) |
| 18 | 0.0641 | 0.00461 (0.0014) | 0.00160 (0.00037) |
| 19 | 0.0652 | 0.00409 (0.0014) | 0.00181 (0.00039) |
| 20 | 0.0581 | 0.00343 (0.0014) | 0.00230 (0.00040) |
| 21 | 0.0346 | 0.00326 (0.0014) | 0.00202 (0.00039) |
| 22 | 0.0714 | 0.00232 (0.0014) | 0.00215 (0.00040) |
| 23 | 0.0430 | 0.00279 (0.0014) | 0.00184 (0.00038) |
| 24 | 0.0634 | 0.00304 (0.0014) | 0.00253 (0.00041) |

The lower limit $(\hat{\delta}_m^l)$ is the derived lower limit of a potentially reversed effect for each month. Estimates are the estimated association (standard errors in parentheses) between mortality and metastases and age, respectively.

minimum, we cannot rule out the existence of an 'unobservable' that could reverse the inference. If, on the other hand, the estimated association between mortality and these two variables is smaller than the derived minimum, we believe it is safe to rule out a reverse effect. The reason for this is that both age and metastases are highly associated with mortality, and, at the same time, the two covariates are imbalanced in the *unmatched sample* (see S2 Table). It is hard to imagine an 'unobservable' being more imbalanced and more associated with mortality than these two covariates.

The result from this analysis is shown in Table 2. The derived lower limit of a potential reverse effect is given in column (2). An association between an unobserved confounder and mortality larger than the derived lower limit, could yield a statistically significant negative effect on mortality from NHT compared to SoC; in other words, a statistically significant reduction in mortality on NHT compared to SoC. The estimated association (standard errors in parentheses) of mortality with age and metastases are presented in columns 3 and 4, respectively. The only larger estimate is in the first month for metastases (see column 4). Thus, the inference could potentially be changed but only for this month. The lower limit is substantially larger than the estimates for all other comparisons.

At a very late stage, we were informed that the grading making up the Gleason score was changed in November 2014 (see, e.g., Epstein, [24]). This means that the Gleason score of the SoC patients is graded according to the old system, whereas some of the NHT patients are graded according to the updated grading system. We, therefore, need to evaluate whether or not the modification is the main reason for the imbalance seen in Fig 4B.

The result from this analysis indicates that the imbalance is not only a consequence of the modification. For details on this evaluation, see S3 Text.

## Discussion

The main result from the paper is that we can rule out a reduction in mortality and skeleton-related events for patients if prescribed NHT rather than being given standard-of-care treatment (SoC) (cf. described in Jonéus et al., [10]. The SoC for this group of patients is, in general, chemotherapy with docetaxel according to a qualitative study in which specialist doctors were asked about their prescription practice (described in Johansson et al. [25]). The results on mortality are, to some extent, in agreement with the results of Chowdhury et al. [9], who found no statistically significant difference in mortality three years after treatment for patients on NHT against those on docetaxel. However, in Chowdhury et al. [9], patients on docetaxel are less healthy at baseline than those on NHT. It is unclear how they adjust for the differences in health at baseline in their analyses.

As we can rule out positive effects on both mortality and do not find any effects on morbidity outcomes, it is difficult to justify the high extra treatment cost of NHT. Rough estimates indicate that the cost for SoC would amount to 400 to 700 Euro for the 6–10 cycles of docetaxel, whereas the cost for NHT would vary between 25,000 and 33,000 Euro during this same period. On the other hand, as docetaxel is given by injection, the cost to the healthcare system is reduced with NHT. Therefore, these costs should also be estimated to make a more relevant comparison of the costs of the two drugs.

The results are obtained when these drugs were made available for a population of men with prostate cancer. As a consequence of using historical controls, the population is restricted to patients prescribed NHT relatively shortly after diagnosis. Potentially, the effect may differ for other populations. There could, of course, be positive effects on outcomes not examined in this study. One such variable could be quality of life. Thus a full-blown cost-effectiveness evaluation needs results from more studies, including studies on outcomes not examined in the paper.

The strength of the study is the access to high-quality register data for a large population of prostate cancer patients. A further important aspect of the study is the pre-published analysis plan, as this allowed for an objective analysis of the data and in conducting sensitivity analyses using data from the quality registers. The main weakness of this study is that only one of the treatments considered as the 1st line treatment for mCRPC is registered. For this reason, it would be valuable to examine if the result could be replicated in a prospective observational study in which docetaxel treatment was captured. In such a prospective study, we recommend to capture potential confounders not available in the registries, e.g. PSA levels at the time of initiating treatment with docetaxel and NHT. We would also recommend to capture the Gleason Score and different metastases as well as the patients' symptoms such as pain and fatigue. We also recommend that not only effectiveness differences (mortality, SRE, and pain) are captured, but also tolerance differences (different side-effects). This is important as the clinicians emphasize in our interviews that docetaxel is more effective, but worse tolerated. For that reason, a comprehensive evaluation of docetaxel also needs to consider aspects related to the patients' tolerance of their treatment.

## Supporting information

**S1 Fig. Distribution of time between diagnosis and NHT prescription.**
(TIF)

**S2 Fig. The effect on mortality of NHT against SoC.** Estimates, 95% Bonferroni confidence intervals and overall mortality for each month (x). Updated entropy weights.
(TIF)

**S3 Fig. The effect on PAIN of NHT against SoC.** Estimates, 95% Bonferroni confidence intervals and overall level of PAIN for each month (x). The effects are all very small and for the majority of months there are no individuals in the treatment group with prevalence of PAIN.
(TIF)

**S4 Fig. Effects on PAIN of NHT against SoC.** Estimates and 95% Bonferroni corrected confidence intervals. We let all patients who die either have no morbidity outcome or a morbidity outcome (i.e. PAIN = 0 or PAIN = 1). Since the mortality with the NHT is observed to be higher than without a SoC the first case (i.e. PAIN = 0) provides a lower bound estimate of the effectiveness of the NHT while the second one provides an upper bound on PAIN. Lower (left panel) and upper (right panel) bounds of potential effect.
(TIF)

**S5 Fig. The effects on mortality of NHT against SoC.** Estimates, 95% Bonferoni corrected confidence intervals and overall level of mortality for each month. Early (left panel) and late (right panel) prescriptions.
(TIF)

**S6 Fig. The effects on SRE of NHT against SoC.** Estimates, 95% Bonferoni corrected confidence intervals and overall level of SRE for each month. Early (left panel) and late (right panel) prescriptions.
(TIF)

**S7 Fig. The effects on PAIN of NHT against SoC.** Estimates, 95% Bonferoni corrected confidence intervals and overall level of SRE for each month. Early (left panel) and late (right panel) prescriptions.
(TIF)

**S8 Fig. The effect on SRE of NHT against SoC.** Estimates, 95% Bonferroni confidence intervals and overall mortality for each month. Updated entropy weights.
(TIF)

**S9 Fig. The effect on PAIN of NHT against SoC.** Estimates, 95% Bonferroni confidence intervals and overall mortality for each month. Updated entropy weights.
(TIF)

**S10 Fig. Results from the placebo regressions.** Estimate, 95% Bonferroni corrected confidence and overall level of SPSA and Gleason for each month. SPSA (left) and Gleason (right). Updated entropy weights.
(TIF)

**S11 Fig. The effect on mortality of NHT against SoC.** Estimates, 95% Bonferroni confidence intervals and overall mortality for each month. Entropy weights from including GleasSa.
(TIF)

**S1 Table. Results from a discrete time Cox regression.**
(DOCX)

**S2 Table. Summary statistics.** Means (standard deviations).
(DOCX)

**S1 Text. Analysis of missing in the proxy variables.**
(DOCX)

**S2 Text. Inference under confounding.**
(DOCX)

**S3 Text. New grading system for Gleason score.**
(DOCX)

## Acknowledgments

The authors thank seminar participants at Uppsala University for their valuable comments and the Swedish Dental and Pharmaceutical Benefits Agency (TLV) for valuable funding.

## Author Contributions

**Conceptualization:** Per Johansson, Sophie Langenskiöld.

**Formal analysis:** Paulina Jonéus, Per Johansson.

**Funding acquisition:** Sophie Langenskiöld.

**Methodology:** Per Johansson, Sophie Langenskiöld.

**Project administration:** Sophie Langenskiöld.

**Software:** Paulina Jonéus.

**Supervision:** Per Johansson.

**Writing – original draft:** Per Johansson.

**Writing – review & editing:** Paulina Jonéus, Per Johansson, Sophie Langenskiöld.

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
