## [Decision Letter · Decision Letter 0]

13 Jun 2023

PONE-D-23-09599Novel hormonal therapy versus Standard of Care -- a registry-based comparative effectiveness evaluation for mCRPC-patientsPLOS ONE

Dear Dr. Langenskiöld,

Thank you for submitting your manuscript to PLOS ONE. After careful consideration, we feel that it has merit but does not fully meet PLOS ONE’s publication criteria as it currently stands. Therefore, we invite you to submit a revised version of the manuscript that addresses the points raised during the review process.

We look forward to receiving your revised manuscript.

Kind regards,

Keiko Hosohata, Ph.D.

Academic Editor

PLOS ONE

Journal Requirements:

4. Please amend the manuscript submission data (via Edit Submission) to include author Per Johansson and Paulina Jonéus.

Reviewers' comments:

Reviewer's Responses to Questions

**Comments to the Author**

1. Is the manuscript technically sound, and do the data support the conclusions?

Reviewer #1: Yes

Reviewer #2: Yes

Reviewer #3: Yes

Reviewer #4: Yes

Reviewer #5: Partly

Reviewer #6: Yes

Reviewer #7: Yes

2. Has the statistical analysis been performed appropriately and rigorously? 

Reviewer #1: Yes

Reviewer #2: Yes

Reviewer #3: Yes

Reviewer #4: Yes

Reviewer #5: Yes

Reviewer #6: Yes

Reviewer #7: I Don't Know

3. Have the authors made all data underlying the findings in their manuscript fully available?

Reviewer #1: Yes

Reviewer #2: Yes

Reviewer #3: Yes

Reviewer #4: Yes

Reviewer #5: Yes

Reviewer #6: Yes

Reviewer #7: Yes

4. Is the manuscript presented in an intelligible fashion and written in standard English?

Reviewer #1: Yes

Reviewer #2: Yes

Reviewer #3: Yes

Reviewer #4: Yes

Reviewer #5: No

Reviewer #6: Yes

Reviewer #7: Yes

5. Review Comments to the Author

Reviewer #1: Dear Authors

Your submission is a well-thought out piece of writing and follows the guidelines. your submission showed great writing skills. You show a great understanding in your content, create fantastic imagery, and evoke many complex emotions in your writing.

Reviewer #2: Manuscript is written in a compressive way. All the analysis and data presented in the manuscript is sufficient to support the said results. It will be a good contribution for further studied on this topic

Reviewer #3: Review Comments: Novel hormonal therapy versus Standard of Care -- a registry2 based comparative effectiveness evaluation for mCRPC-patients

Generally, the manuscript titled "Novel hormonal therapy versus Standard of Care - a registry-based comparative effectiveness evaluation for mCRPC patients" provides valuable insights into the comparative effectiveness of Novel Hormonal Therapy (NHT) versus Standard of Care (SoC) for patients with metastatic castration-resistant prostate cancer (mCRPC). The study objectives and methodology are generally well-described, but there are several areas that require further clarification and improvement. Below, I provide specific comments and suggestions for each section of the manuscript.

Abstract:

The abstract provides a clear understanding of the study objectives and methodology. there are a few areas that could benefit from further clarification and improvement. Specifically:

Line 14: The statement "one of the few comparative effectiveness evaluations" needs to be supported by evidence. It would be helpful to mention any existing studies that have evaluated the comparative effectiveness of NHT and SoC in mCRPC.

Line 28: The abstract states that there is a substantial increase in mortality for patients prescribed with NHT, but it would be helpful to quantify this increase and provide a measure of effect size or a confidence interval.

Line 32: The statement "We can also rule out a positive effect on skeleton-related events (SRE) for the NHT patients" should be supported by specific findings or statistical results.

Introduction:

The introduction provides an overview of the study objectives and methodology. However, there are a few areas that could benefit from further clarification and improvement. Specifically:

Line 47: The statement that "there is only one study comparing chemotherapy and NHT" should be supported by a citation to validate this claim.

Line 49: The authors mention differences in baseline characteristics between patients on docetaxel and NHT but do not clearly state whether these differences were adjusted for in their analyses. This should be clarified.

Methods:

The methods section is well-structured and provides a comprehensive overview of the research design. However, there are a few areas that require clarification and improvement. Specifically:

1. The authors should consider expanding the methods section to provide more context and justification for the chosen entropy balancing methodology. This would enhance the understanding of readers who may not be familiar with the statistical techniques.

2. The rationale behind the selection of a 36-month window for NHT-treated patients and the choice of June 2008 to June 2010 for the comparison group should be explained more explicitly.

3. The authors should provide a more detailed explanation of the Elixhouser comorbidity index and its relevance to the study. Additionally, including references for the mentioned studies would be helpful.

Results:

The results section is well-structured and presents the findings clearly. However, there are a few points that could be addressed for further clarity. Specifically:

1. The analysis of skeleton-related events (SRE) and severe pain reveals interesting findings. The higher proportion of SRE in the NHT group suggests a potential association with the treatment. However, due to the higher mortality in the NHT group, the estimation of effects requires bounds to account for observation bias. The authors adequately address this limitation.

2. The analysis of early and late prescriptions of NHT indicates a potential difference in the effects depending on the timing of prescription. However, the overlapping confidence intervals and the lack of a clear pattern prevent firm conclusions. This finding highlights the need for further investigation and does not allow for strong generalizations.

3. The discussion of missing data and the assessment of missingness are crucial for understanding the limitations of the study. By treating missing observations as missing at random, the authors address this issue appropriately. However, the impact of missing data on the results should be acknowledged and discussed further.

4. The analysis of potential confounders, such as age and metastases, provides insights into the robustness of the mortality estimates. The derived lower limits and comparisons with known covariates help assess the potential for reversed effects. The authors appropriately discuss the limitations and implications of these findings.

5. The article would benefit from providing a summary of the main findings at the end of the results section. This would allow readers to quickly grasp the key outcomes without having to revisit the entire section.

Discussion:

Generally, discussion section of the article provides a comprehensive analysis of the main results and their implications. However, there are a few areas where additional clarification and expansion would be beneficial. Below are specific comments and suggestions for improvement.

1. In paragraph 1, it would be helpful to provide more context regarding the study by Johansson et al. (2021) and its relevance to the current research.

2. The authors discuss the potential reasons behind the higher mortality in the NHT group, mentioning the lack of clinical trial evidence and concerns regarding the potential long-term side effects of NHT. Expanding on these potential reasons and discussing alternative hypotheses would contribute to the understanding of the findings.

3. The authors mention the strengths of the study, such as the large sample size and the use of registry data. However, it would be beneficial to provide a more comprehensive overview of the study's strengths and limitations in a dedicated paragraph.

4. It would be beneficial to include a brief discussion of the implications of the results for clinical practice or future research. Additionally, highlighting the main limitations and potential areas for further investigation would contribute to the completeness.

Generally, this study provides valuable insights into the comparative effectiveness of NHT versus SoC for mCRPC patients. Addressing the points mentioned above will significantly strengthen the manuscript and enhance its contribution to the field.

Reviewer #4: the tittle should be revised to provide more clarity. the introduction provides the clear overview of the problem emphasizing their potential impact of therapy to the patients. the references are also relevant to the context of the research. the methods are also structured and provide the complete details of data collection settings and study design. this scientific article is well written and makes valuable contribution to the field.

Reviewer #5: 1. It was stated that prostate cancer is the fifth leading cause of death….it would be interesting to have the statistics to back the claim

2. In the introductory part, some reports are been presented and this calls for a reorganization

3. The authors need to recast the introduction part to bring out the gap clearly

4. Authors are to move the approval of the study to the methods part

5. Some results were presented in the methods, authors are to review this

6. Authors are to state clearly which of the software was used for post-analysis tests.

7. The discussion part was not properly written and was too scanty, the 1st paragraph should be about your findings before relating it. kindly update this important part

8. The authors citing method is somehow strange, for example “for more details, see…”. Authors are to check the professional way of citation in the previous article of PLUS ONE

9. The whole manuscript needs to be checked for grammatical errors.

10. The conclusion part was missing

Reviewer #6: The topic is interesting and presents the quality of the work. The topic is interesting and presents the quality of the work. The work highlighted the reduction in mortality and skeleton related events for patients if prescribed NHT rather than being given standard-of-care treatment (SoC). The SoC in general, chemotherapy with docetaxel the mortality are to some extent statistically significant difference in mortality three years after the treatment for patients on NHT against those on docetaxel.

Reviewer #7: Overall , the amnuscript is good in its presentauion, whereas there are a few typographical and sentence steructre errors which must be rectifoed by the authors. Besides, this the comments send to the editor should be also addressed.

6. PLOS authors have the option to publish the peer review history of their article (what does this mean?). If published, this will include your full peer review and any attached files.

Reviewer #1: No

Reviewer #2: No

Reviewer #3: **Yes: **Dr. Zeeshan AHMED (PhD, MS, MBA, Pharm.D)

Reviewer #4: No

Reviewer #5: **Yes: **Clement Olusoji Ajayi

Reviewer #6: **Yes: **Dr. Abrar Ahmed

Reviewer #7: **Yes: **Dr. Sheryar Afzal

---

## [Author Response · Author response to Decision Letter 0]

11 Aug 2023

Dear Reviewer#1: 

We first of all thank you for taking the time to review our manuscript. The manuscript is not only long but also complex, so we appreciate the investment you have made in it. 

Secondly, we are pleased that our submission follow the guidelines, show great writing skills, and influence your emotions. We are also pleased that we show an understanding for the topic. 

Finally, we hope that you will enjoy the revised version even further. We have made a lot effort in order to clarify complex issues in a more intuitive way.

Dear Reviewer #2: 

We first of all thank you for taking the time to review our manuscript. The manuscript is not only long but also complex, so we appreciate the investment you have made in it. 

Secondly, we are happy that you believe our manuscript is written in a comprehensive way, and also that the analysis support the results. We are also pleased that you think our paper 

would be a good start for further research in the field. 

Finally, we hope that you will enjoy the revised version even further. We have made a lot effort in order to clarify complex issues in a more intuitive way.

Dear Dr. Zeeshan AHMED (Reviewer #3)

We first of all thank you for all the time you have taken to review our manuscript. From the comments you have given, we understand that you have spend hours/days with out paper.

Also, your comments were are well-thought, so we have addressed all of them. We believe your comments have raised the quality of the paper substaintially. 

We summarize the way we addressed your comment below. As you did, we have referred to the line for the changes, so you hopefully easily can see the changes in their context. 

Your comment #1: Line 14: The statement "one of the few comparative effectiveness evaluations" needs to be supported by evidence. It would be helpful to mention any existing studies that have evaluated the comparative effectiveness of NHT and SoC in mCRPC

Our response to #1 : In the manuscript (see line 44-49, we have now summarized the two previous comparative effectiveness studies that we are aware of which study mortality differences. 

Your comment #2: Line 28: The abstract states that there is a substantial increase in mortality for patients prescribed with NHT, but it would be helpful to quantify this increase and provide a measure of effect size or a confidence interval.

Our response to #2: We are evaluating the effects for many periods, and these effects cannot 

straightforwardly be translated to a single magnitude together with a correct confidence interval. We also prefer the non-parametric hazard model estimated with fewer assumptions, which do not provide the point estimated provided by a proportional hazard model. However, we now have added 64% as a point estimate which is the average increase in mortality over the 24 months evaluated at the overall mean mortality for the two groups (see lines 27 and 265).

Your comment #3: Line 32: The statement "We can also rule out a positive effect on skeleton-related events (SRE) for the NHT patients" should be supported by specific findings or statistical results.

Our response to #3: Line 32 is revised and reframed (see lines 31-33). The statement is based on the finding in Wong et al. (2019) that most mCRPC patients who die suffer from bone metastases. It is clear that if all mCRPC patients suffer from bone metastases, we can rule out a positive effect from NHT from our result presented in our figure 3. 

However the statement is too strong and we apologise. As a consequence we removed line 295 in the document were we incorrectly wrote “This result allows us to rule out a positive effect on SRE for the NHT patients if they had been prescribed SoC instead.’ 

In addition, we changed the wording in the concluding section (lines 435-436) and discussed the result in a new summary section suggested below (see lines 322-328).

Your comment #4: Line 47: The statement that "there is only one study comparing chemotherapy and NHTa" should be supported by a citation to validate this claim.

Response to #4: At the time of initiating this research, a structured literature review was conducted Only one study was found evaluation chemotherapy against NHT in terms of survival difference. Since then, we became aware of a second study comparing the same treatments. Therefore, we conclude that the two interventions which are claimed to be the alternative 1st line treatment for patients with mCRPC have only been rarely evaluated. We have clarified this in the revised manuscript (se lines 44-49). 

However as there is always a possibility of missing relevant literature, and then especially grey literature we stated “to our knowledge”. We understand that this can be misinterpreted as not having conduced a structured review. In the revised document, it is clarified that a structured literature review was conducted (line 44). 

Your comment #5: Line 49: The authors mention differences in baseline characteristics between patients on docetaxel and NHT but do not clearly state whether these differences were adjusted for in their analyses. This should be clarified.

Our response to #5: The statement on this study refers to Chowdhury et al., (2020) and not our study. As a consequence line 49-51 are revised (see lines 47-49).

Your comment #6: The authors should consider expanding the methods section to provide more context and justification for the chosen entropy balancing methodology. This would enhance the understanding of readers who may not be familiar with the statistical techniques.

Our response to #6: The method is explained in detail in the pre-analysis Jonéus et al. (2021). For this reason, we did not provide any details on intuition for the method in this paper. We apologize. In this version we have stressed to provide an intuition by comparing it to propensity score matching (see lines 192-202).

Your comment #7: The rationale behind the selection of a 36-month window for NHT-treated patients and the choice of June 2008 to June 2010 for the comparison group should be explained more explicitly.

Our response to #7: As in general, timing of treatment is not acknowledged in effectiveness analysis we understand that the sampling scheme is hard to grasp. Details (including figures) on the sampling is provided in Jonéus et al. (2021). However, we understand that the writing was to scant. We now included more than half a page where we provided the intuition for the sampling procedure (see lines 117-146). 

Your comment #8: The authors should provide a more detailed explanation of the Elixhouser comorbidity index and its relevance to the study. Additionally, including references for the mentioned studies would be helpful.

Our response to #8: The present version provides a reference to (Lieffers et al., 2011) who compares Charlson and Elixhauser comorbidity measures to predict colorectal cancer survival using administrative health data (see lines 155-158). They state that the Elixhauser index is a superior comorbidity risk-adjustment method.

Your comment #9: The analysis of skeleton-related events (SRE) and severe pain reveals interesting findings. The higher proportion of SRE in the NHT group suggests a potential association with the treatment. However, due to the higher mortality in the NHT group, the estimation of effects requires bounds to account for observation bias. The authors adequately address this limitation.

Our response to #9: We do not find any critique in this paragraph, why we did not make any revision.

Your comment #10: The analysis of early and late prescriptions of NHT indicates a potential difference in the effects depending on the timing of prescription. However, the overlapping confidence intervals and the lack of a clear pattern prevent firm conclusions. This finding highlights the need for further investigation and does not allow for strong generalizations.

Our response to #10: As we do not make any strong generalization, we do not find any critique in this paragraph. For that reason, we did not make any revision.

Your comment #11: The discussion of missing data and the assessment of missingness are crucial for understanding the limitations of the study. By treating missing observations as missing at random, the authors address this issue appropriately. However, the impact of missing data on the results should be acknowledged and discussed further.

Our response to #11: We added two sentences (see lines 350-352) as disclaimer for the possibility that the missing data may be a problem for the sensitivity analysis.

Your comment #12: The analysis of potential confounders, such as age and metastases, provides insights into the robustness of the mortality estimates. The derived lower limits and comparisons with known covariates help assess the potential for reversed effects. The authors appropriately discuss the limitations and implications of these findings.

Our response to #12: We do not find any critique in this paragraph, why we did not make any revision.

Your comment #13: The article would benefit from providing a summary of the main findings at the end of the results section. This would allow readers to quickly grasp the key outcomes without having to revisit the entire section.

Our response to #13: We have done so (see lines 322-328).

Your comment #14: In paragraph 1, it would be helpful to provide more context regarding the study by Johansson et al. (2021) and its relevance to the current research.

Our response to #14: Johansson et al. (2021) is the pre-analysis plan for the present study. As a part or the protocol we conducted a qualitative study on doctors prescribing the drugs examined in this study. This is clarified in the present version of the paper (see lines 428-430)

Your comment #15: The authors discuss the potential reasons behind the higher mortality in the NHT group, mentioning the lack of clinical trial evidence and concerns regarding the potential long-term side effects of NHT. Expanding on these potential reasons and discussing alternative hypotheses would contribute to the understanding of the findings.

Our response to #15: We do not know how to address this comment. The reason is that we do not have this discussion in the Discussion section nor in the rest of the document. We also would like to avoid speculation about possible reasons for the mortality difference. The clinicians with whom we conducted the qualitative yet unpublished research emphasize that Docetaxel is more effective than NHT, but the tolerance is worse. These opinions need to be verified in future comparative effectiveness evaluations in which not only effectiveness differences is captured but also side-effect differences. 

Your comment #16: The authors mention the strengths of the study, such as the large sample size and the use of registry data. However, it would be beneficial to provide a more comprehensive overview of the study's strengths and limitations in a dedicated paragraph.

Our response to #16: We struggled with this comment. The reason is that we do have a discussion on the strength of the study in this section. However, we added a paragraph on strengths and weakness, by combing comment 3 and 4, at the end in the Discussion section (see lines 449-463)

Your comment #17: It would be beneficial to include a brief discussion of the implications of the results for clinical practice or future research. Additionally, highlighting the main limitations and potential areas for further investigation would contribute to the completeness.

Our response to #17: We have added a paragraph at the end in the Discussion section (see lines 449-463).

Dear Reviewer # 4

We first of all thank you for taking the time to review our manuscript. The manuscript is not only long but also complex, so we appreciate the investment you have made in it. 

We are also grateful that you find the introduction clear, the references relevant, and the methods structured, and believe the manuscript to contribute to the field. 

We have also considered adjusting the title as you suggest. After careful consideration, however, we have decided to keep the current title, as we can’t find a better one equally short. 

Finally, we hope that you will enjoy the revised version even further. We have made a lot effort in order to clarify complex issues in a more intuitive way.

Dear Dr Clement Olusoji Ajayi (Reviewer #5)

We first of all thank you for taking the time to review our manuscript. The manuscript is not only long but also complex, so we appreciate the investment you have made in it. 

We are grateful that you highlighted (comment #1) that we need a reference for prostate cancer being the fifth leading cause of cancer death. We understand your concern as we by mistake left out that we referred to cancer death, not death in general. 

Also, we clarify the gap in the literature that you believe is lacking in the document (comment #2) by summarizing the structed literature review we did at the time of initiating this research. Only one study was found at that time evaluating chemotherapy against NHT in terms of survival difference. 

Since then, we have become aware of a second study comparing the same treatments. Therefore, we conclude in the manuscript that the two interventions which are claimed to be the alternative 1st line treatment for patients with mCRPC have only been rarely evaluated. We have clarified this in the revised manuscript (se lines 44-49). 

However as there is always a possibility of missing relevant literature, and then especially grey literature we stated “to our knowledge”. We understand that this can be misinterpreted as not having conduced a structured review. In the revised document, it is clarified that a structured literature review was conducted (line 44). 

We do agree with you that the approval of the study can be moved to the method part (comment #4) and has done so in the current version of the paper. 

We are grateful that you saw that we had forgotten to specify the software for our analyses (comment #6). We have done so in the current version of the paper.

We have also asked professionals for checking the manuscript for grammatical errors, which have been done in the current version. 

We also agree with you that the conclusion part of the manuscript is missing (comment #10). For that reason, we have summarized our results under separate section. 

We have also changed the way we refer to figure and tables in the manuscript and also to the supplementary information. Thank you for emphasizing this lack of compliance. 

However, we do not present any result in the method section as you state (comment #5). The method section only discusses descriptive statistics of the sample at baseline. 

Also, after careful consideration we have decided not to change the discussion part of the paper (comment 7), as our other six reviewers thought it was well-written. 

With the current changes, we hope that you will understand the revised version better. We have also made a lot other changes to alleviate the understanding of the paper. 

Dear Dr. Abrar Ahmed (Reviewer #6)

We first of all thank you for taking the time to review our manuscript. The manuscript is not only long but also complex, so we appreciate the investment you have made in it. 

Secondly, we are pleased that you do not seem to request any changes to the sent-in version of the document.

Finally, we hope that you will enjoy the revised version even further. We have made a lot effort in order to clarify complex issues in a more intuitive way.

Dear Dr. Sheryar Afzalm (Reviewer #7)

We first of all thank you for taking the time to review our manuscript. The manuscript is not only long but also complex, so we appreciate the investment you have made in it. 

Secondly, we are also grateful that you only found few typographical and structural errors. The language and the grammar have now been checked professionally. 

Finally, we hope that you will enjoy the revised version even further. We have made a lot effort in order to clarify complex issues in a more intuitive way.

---

## [Editor Report · Decision Letter 1]

16 Aug 2023

Novel hormonal therapy versus Standard of Care -- a registry-based comparative effectiveness evaluation for mCRPC-patients

PONE-D-23-09599R1

Dear Dr. Langenskiöld,

We’re pleased to inform you that your manuscript has been judged scientifically suitable for publication and will be formally accepted for publication once it meets all outstanding technical requirements.

Kind regards,

Keiko Hosohata, Ph.D.

Academic Editor

PLOS ONE
---

## [Editor Report · Acceptance letter]

25 Aug 2023

PONE-D-23-09599R1 

Novel hormonal therapy versus standard of care -- A registry-based comparative effectiveness evaluation for mCRPC-patients 

Dear Dr. Langenskiöld:

I'm pleased to inform you that your manuscript has been deemed suitable for publication in PLOS ONE. Congratulations! Your manuscript is now with our production department. 

Kind regards, 

on behalf of

Dr Keiko Hosohata 

Academic Editor

PLOS ONE